# Impact of COPD case finding on clinical care: a prospective analysis of the TargetCOPD trial

Shamil Haroon [1], Peymane Adab [1], Andrew P Dickens [1], Alice J Sitch [1], Kiran Rai [1], Alexandra Enocson,[1] David A Fitzmaurice,[2] Rachel E Jordan[1]

► Prepublication history and supplemental material for this paper are available online. To view these files, please visit the journal online (http://dx.doi.org/10.1136/bmjopen-2020-038286).

¹Institute of Applied Health Research, University of Birmingham, Birmingham, UK
²Warwick Medical School - Health Sciences, University of Warwick, Coventry, UK

**Correspondence to**
Dr Shamil Haroon;
s.haroon@bham.ac.uk

## ABSTRACT

**Objectives**  To investigate the impact of chronic obstructive pulmonary disease (COPD) case finding on clinical care.

**Design**  We conducted a prospective observational analysis of data from a pragmatic cluster randomised controlled trial in primary care in the West Midlands, UK (TargetCOPD). This compared alternative methods of COPD case finding against usual care. Data were extracted from electronic healthcare records and self-reported questionnaires for a subset of patients with newly diagnosed COPD.

**Setting**  50 general practices that participated in the TargetCOPD trial.

**Participants**  Patients aged 40–79 years newly identified with COPD by targeted case finding or by usual care, from 10 August 2012 to 22 June 2014.

**Primary and secondary outcome measures**  The primary outcome was addition to a COPD register by the end of the trial. The secondary outcome was a clinical care score, derived from the sum of clinical assessments and relevant interventions. Associations between participant characteristics and the primary and secondary outcomes were assessed using multilevel regression.

**Results**  857 patients identified with COPD by case finding and 764 by usual care were included. Only 21.2% of case-found patients had been added to a COPD register, compared with 92.7% of those diagnosed by usual care. The odds of being added were greater in smokers (adjusted OR 8.68, 95% CI 2.53 to 29.8), and in those with lower percentage of predicted forced expiratory volume in 1 s (adjusted OR 0.96 per percentage rise, 95% CI 0.95 to 0.98). Patients who had been added to a COPD register had a significantly higher clinical care score (mean difference 5.06, 95% CI 4.36 to 5.75).

**Conclusions**  Only one in five case-found patients had been registered with COPD. Patients added to a COPD register received significantly higher levels of appropriate clinical care.

**Trial registration number**  ISRCTN14930255; Post-results.

### Strengths and limitations of this study

► We conducted a detailed evaluation of the respiratory care received by patients with case-found chronic obstructive pulmonary disease and compared this to that received by patients diagnosed by usual care over the same period.
► The analyses adjusted for a number of important confounding factors, including measures of disease severity.
► The analysis was done on only a subset of trial participants.
► Electronic healthcare record data quality was highly dependent on clinical coding practices.
► The clinical care score represents a relatively crude estimate of overall respiratory care and individual components of the score were not weighted for their relative importance.

COPD globally and a number of experts and policy makers have called for early detection and screening.[2] This has the potential to prevent a significant burden of premature morbidity and mortality. A targeted case finding approach in England could reduce COPD hospitalisations by an estimated 3300 per year and prevent almost 3000 premature deaths over 3 years.[3] A recent model-based evaluation also concluded that systematic case finding for COPD could be cost-effective in the long term.[4] However, this is based on the assumption that case-found patients go on to receive improved clinical care.

The UK National Screening Committee (NSC) requires evidence from high quality randomised controlled trials (RCTs) that a screening programme reduces mortality or morbidity and that the benefits outweigh any harms, before recommending population-based screening.[5] A systematic review for the US Preventive Services Task Force concluded that screening asymptomatic people for COPD could not be recommended because of a lack of evidence that it improves health-related quality of life, morbidity or mortality.[6]

## INTRODUCTION

Chronic obstructive pulmonary disease (COPD) is the third leading cause of death worldwide and a major cause of disability.[1] There is a large burden of undiagnosed

However, this does not necessarily apply to patients with symptoms.[7]

A large number of studies have evaluated case finding approaches for COPD in primary care.[8 9] However, few have assessed the clinical outcomes of case-found patients compared with those diagnosed through usual care. TargetCOPD was a large cluster RCT in the West Midlands, UK which confirmed the effectiveness and cost-effectiveness of a targeted programme to identify new COPD cases in the primary care setting compared with usual care.[10]

Follow-up data were collected on participants identified with COPD during the trial period to assess the clinical interventions they subsequently received. The objective of this study was to describe the clinical care and management of patients newly identified with COPD by targeted case finding and compare this with those diagnosed by usual care over the same period. In addition, we assessed which patient and practice-level factors were associated with better management of COPD among those who were newly diagnosed.

## METHODS
### Study design

We undertook a prospective observational analysis of data on the clinical care of patients newly diagnosed with COPD in the TargetCOPD trial.[10] TargetCOPD was a cluster RCT comparing the effectiveness and cost-effectiveness of alternative approaches to targeted case finding against usual care on the identification of previously undiagnosed COPD.

Patients aged 40–79 years with no prior diagnosis of COPD were identified from electronic general practice registers. Those in the case finding arm were provided screening questionnaires by post and/or opportunistically when visiting their practice. Patients reporting respiratory symptoms were then invited for a spirometry assessment. Post-bronchodilator spirometry was performed by trained researchers according to American Thoracic Society and European Respiratory Society 2005 guidelines[11] using an ultrasonic flow head spirometer (Spiroson-AS, ndd Medical Technologies, Zurich, Switzerland).

The study team sent letters to general practitioners (GPs) informing them of patients identified as likely to have COPD through case-finding (see later for definition) with advice to follow the relevant National Institute for Health and Care Excellence (NICE) guidelines.[12] Patients newly diagnosed either by case finding or by usual care during the trial period were tracked through GP records. The trial was active in each practice for a period of 1 year, with a staggered start from 10 August 2012 to 22 June 2013.

From 6 October 2015 to 12 October 2016, data on the clinical care of COPD were extracted from the electronic healthcare records (EHRs) of a sample of eligible patients from both arms of the trial (clinical codes listed in online supplemental table 1). All case-found patients who agreed to further study were sent questionnaires from 30 March to 11 April 2018 that included detailed questions about the management of their COPD.

### Setting

This analysis used data from 50 out of 54 general practices that participated in the TargetCOPD trial. Relevant data could not be extracted from four practices due to practice closures (n=3) and missing practice lists (n=1).

### Participants

In this study we included those aged 40–79 years at baseline who were newly identified with COPD by targeted case finding or by usual care, from 10 August 2012 to 22 June 2014. Case-found COPD was defined as a post-bronchodilator forced expiratory volume in 1 s ($FEV_1$) to forced vital capacity ratio of less than 0.7 (in line with recommended UK guidelines at the time) in those who reported respiratory symptoms (chronic cough or phlegm for at least 3 months of the year for two or more years, wheeze in the previous 12 months or Medical Research Council (MRC) grade 2 dyspnoea or higher). Newly diagnosed COPD by usual care was defined as a clinician diagnosis recorded on the EHR using predefined clinical codes (see online supplemental table 2) made independently of case finding. This included all patients diagnosed with COPD in the usual care arm and those diagnosed in the case finding arm prior to receiving a trial spirometry assessment.

### Outcomes

The primary outcome was the addition of patients to a COPD register by the close of the trial period. There is currently a contractual requirement for general practices in England to maintain a register of patients with COPD.[13] Addition to a register is clinician-led but supported by administrators.

The secondary outcome was a composite clinical care score, derived by summing the number of clinical assessments and interventions recorded (one point for each) from the end of the trial up to 2 years of follow-up. The components of the score were based on clinical assessments and interventions that were relevant to this patient group, based on NICE guidelines.[12] Clinical assessments included recording MRC dyspnoea score, COPD assessment test (CAT), post-bronchodilator spirometry, COPD severity, body mass index (BMI), oxygen saturation, chest X-ray and screening for depression and anxiety. Clinical interventions included recording a care plan, annual review, smoking cessation support (brief advice and nicotine replacement therapy (NRT) for current smokers), influenza and pneumococcal vaccination, referral for pulmonary rehabilitation, assessment of inhaler technique, and prescriptions of inhalers, antibiotics, and prednisolone. Sensitivity analyses included data from the beginning of the trial up to 3 years of follow-up.

## Other variables

Data on demographic characteristics, smoking status and comorbidities (asthma, ischaemic heart disease, heart failure, diabetes, depression, anxiety, osteoporosis and stroke) were also obtained for included participants from the main trial dataset. Practice level data were extracted including patient list size, socioeconomic status (based on the Index of Multiple Deprivation (IMD)), percentage of the registered patient list from non-white ethnicities, and the baseline percentage of patients already diagnosed with COPD (data provided by each practice).

## Statistical methods

The primary analysis used a multilevel logistic regression model to investigate the association between the odds of being added to a COPD register and patient characteristics, among those with case-found COPD. This included the practice as a cluster variable, and demographic and clinical characteristics as independent variables, adjusting for age, sex, smoking status, percentage of predicted $FEV_1$, number of self-reported comorbidities and CAT score. A complete case analysis was performed due to low levels of missing data.

The secondary analysis used a multilevel linear regression model among participants diagnosed with COPD during the trial to assess associations between the clinical care score and patient characteristics, diagnostic route (case-found vs usual care) and addition to a COPD register. The model was run separately using EHR and questionnaire data, with the latter restricted to participants with case-found COPD. Associations between the percentage of case-found patients added to COPD registers and practice characteristics were explored using Pearson correlation coefficients. All analyses were performed using Stata M/P V.14.2.

## Role of the funding source

The study sponsors had no role in the study design; in the collection, analysis and interpretation of data; in the writing of the report; and in the decision to submit the paper for publication. The corresponding authors had full access to all the data in the study and have final responsibility for the decision to submit for publication.

## Patient and public involvement

The Birmingham Lung Improvement Studies (BLISS) Patient Advisory Group contributed to participant recruitment, design of questionnaires and interpretation of data.

## RESULTS

### Participants

Over the trial period, 1621 people with previously undiagnosed COPD were identified in participating practices, including 857 through case finding, and 764 by usual care (337 of these from the usual care arm).[10] EHR data on clinical care were available for 532 patients who were identified with COPD through either case finding or usual care (figure 1). Among the 857 case-found patients, 375 (43.8%) returned questionnaires. The characteristics of participants with and without additional EHR and questionnaire data were broadly similar (see online supplemental table 3).

The mean age of participants newly diagnosed with COPD by case finding and usual care was similar (63.8 and 63.7 years, respectively), a similar proportion were men, and the majority (65.6%) were white British (table 1). A greater proportion of participants diagnosed by usual care were current smokers compared with those diagnosed by case finding (54.6% vs 30.2%), although the proportion with chronic conditions was similar. The subsample with case-found COPD who returned a questionnaire were slightly older, more likely to be men, and generally reported fewer comorbidities than the full sample of case-found patients (see online supplemental table 4).

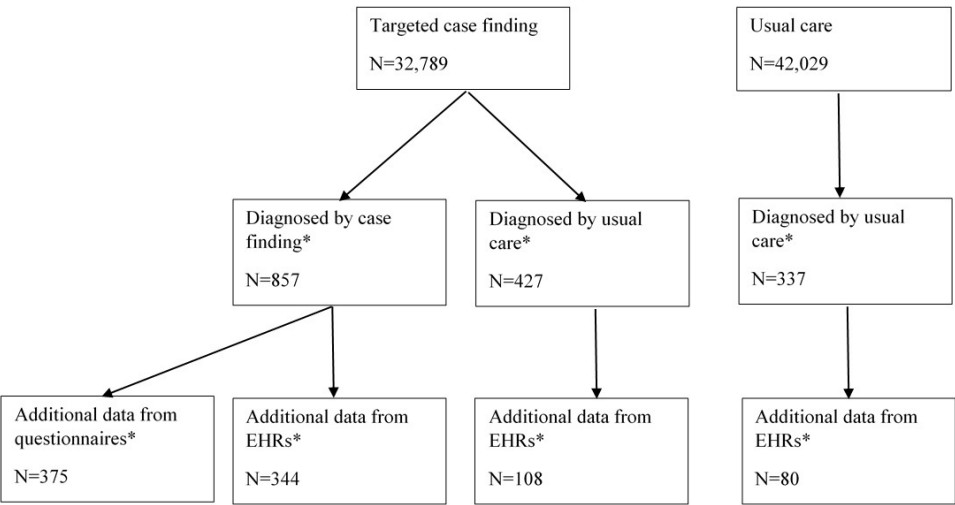

**Figure 1** Participant flow diagram. *Included in the current study. EHR, electronic healthcare record.

**Table 1** Characteristics* of participants newly diagnosed with chronic obstructive pulmonary disease by case-finding and usual care

| | | Diagnostic route | | | |
|---|---|---|---|---|---|
| | | Case-finding (n=857) | | Usual care (n=764) | |
| | | N | (%) | N | (%) |
| Age (years) | Mean (SD) | 63.8 | 9.6 | 63.7 | 9.7 |
| | 40–49 | 91 | 10.6 | 76 | 9.9 |
| | 50–59 | 190 | 22.2 | 184 | 24.1 |
| | 60–69 | 320 | 37.3 | 275 | 36 |
| | 70+ | 256 | 29.9 | 229 | 30 |
| Sex | Male | 489 | 57.1 | 418 | 54.7 |
| Ethnic group | White British | 556 | 64.9 | 507 | 66.4 |
| | Mixed | 5 | 0.6 | 5 | 0.7 |
| | Asian | 28 | 3.3 | 11 | 1.4 |
| | African/ Caribbean | 14 | 1.6 | 24 | 3.1 |
| | Other | 17 | 2 | 3 | 0.4 |
| | Unknown | 237 | 27.7 | 214 | 28 |
| Smoking status | Current | 259 | 30.2 | 417 | 54.6 |
| | Former | 372 | 43.4 | 255 | 33.4 |
| | Never | 124 | 14.5 | 15 | 2 |
| | Unknown | 102 | 11.9 | 77 | 10.1 |
| Comorbidities† | Asthma | 71 | 20.6 | 46 | 24.5 |
| | Ischaemic heart disease | 44 | 12.8 | 26 | 13.8 |
| | Heart failure | 9 | 2.6 | 7 | 3.7 |
| | Diabetes | 58 | 16.9 | 30 | 16 |
| | Depression | 19 | 5.5 | 19 | 10.1 |
| | Anxiety | 10 | 2.9 | 5 | 2.7 |
| | Osteoporosis | 12 | 3.5 | 8 | 4.3 |
| | Stroke | 6 | 1.7 | 6 | 3.2 |
| | No comorbidities | 172 | 50 | 84 | 44.7 |
| | 1 comorbidity | 123 | 35.8 | 71 | 37.8 |
| | ≥2 comorbidities | 49 | 5.7 | 33 | 4.3 |

*Based on data from electronic healthcare records.
†Based on a subset of patients (344 patients diagnosed by case-finding and 188 diagnosed by usual care).

## Practice characteristics

The mean practice list size was 5762 patients (SD 3482). The mean socioeconomic status (mean IMD score 36.7, SD 15.2) was similar to Birmingham as a whole (mean 37.8).[14] The majority of registered patients were white British (mean 80.5%, SD 20.5) and the mean prevalence of diagnosed COPD at the start of the trial was 1.6% (SD 0.6).

## Addition to COPD registers

Of the 857 patients with case-found COPD, only 182 (21.2%) had been added to a COPD register, compared with 708 out of 764 patients (92.7%) diagnosed by usual care. Among those with case-found COPD, the median

**Table 2** Multilevel logistic regression model* assessing the association between participant characteristics and likelihood of being listed on a chronic obstructive pulmonary disease (COPD) register among those with case-found COPD (n=754)

| Participant characteristics | aOR† | (95% CI) | P value |
|---|---|---|---|
| Age | 1.02 | (0.99 to 1.05) | 0.139 |
| Sex (male) | 1.01 | (0.63 to 1.60) | 0.976 |
| Smoking status | | | |
| Never smoked (reference) | | | |
| Ex-smoker | 6.32 | (1.88 to 21.29) | 0.003 |
| Current smoker | 8.68 | (2.53 to 29.82) | 0.001 |
| $FEV_1$ % predicted | 0.96 | (0.95 to 0.98) | <0.001 |
| Self-reported comorbidities (n) | 0.97 | (0.78 to 1.20) | 0.766 |
| CAT score | 1.02 | (0.99 to 1.05) | 0.218 |

*Using data from the main TargetCOPD trial dataset.
†aOR estimated by a multilevel logistic regression model, accounting for clustering by practice and adjusted for all variables listed in the table.
aOR, adjusted OR; CAT, COPD assessment test; $FEV_1$, forced expiratory volume in 1 s.

time from the trial spirometry assessment to being added to a COPD register was 152 days (IQR 72–258).

Among case-found patients, the odds of being added to a COPD register were higher among current or former smokers (adjusted OR 8.68, 95% CI 2.53 to 29.8 and 6.32, 95% CI 1.88 to 21.3, respectively), and those with a lower percentage of predicted $FEV_1$ (adjusted OR 0.96 per percentage rise, 95% CI 0.95 to 0.98; table 2).

The median percentage of case-found patients added to a COPD register in each general practice was 14.5% (IQR 3.2%–32.3%). Overall there were no significant correlations between the percentage of patients added to a COPD register and the measured practice characteristics (table 3).

**Table 3** Correlations between the percentages of case-found patients added to COPD registers and practice characteristics

| Practice characteristics | Correlation coefficient* | P value |
|---|---|---|
| Non-white ethnic groups (%) | 0.15 | 0.45 |
| Patient list size | 0.20 | 0.33 |
| Baseline prevalence of diagnosed COPD | 0.03 | 0.89 |
| Total number of case-found patients | −0.09 | 0.66 |
| Socioeconomic status | 0.33 | 0.09 |

*Pearson's correlation coefficient.
COPD, chronic obstructive pulmonary disease.

**Table 4** Clinical care during the 2-year follow-up of participants with electronic healthcare record data who were newly diagnosed with COPD by case-finding or usual care

| | | Diagnostic route | | | |
| | | Case finding (n=344) | | Usual care (n=188) | |
| | | N* | (%) | N* | (%) |
|---|---|---|---|---|---|
| Clinical assessments | | | | | |
| MRC dyspnoea score recorded | | 98 | 28.5 | 171 | 91 |
| CAT score recorded | | 36 | 10.5 | 94 | 50 |
| Spirometry undertaken | | 48 | 14 | 79 | 42 |
| COPD severity recorded | | 33 | 9.6 | 96 | 51.1 |
| BMI recorded | | 244 | 70.9 | 168 | 89.4 |
| Oxygen saturations recorded | | 41 | 11.9 | 55 | 29.3 |
| Chest X-ray undertaken | | 13 | 3.8 | 9 | 4.8 |
| Depression screen undertaken | | 54 | 15.7 | 55 | 29.3 |
| Clinical interventions | | | | | |
| Listed on COPD register | | 78 | 22.7 | 175 | 93.1 |
| Care plan recorded | | 38 | 11 | 97 | 51.6 |
| Annual review undertaken | | 91 | 26.5 | 170 | 90.4 |
| Smoking cessation counselling | | 157 | 45.6 | 139 | 73.9 |
| Nicotine replacement therapy | | 27 | 7.8 | 17 | 9 |
| Influenza vaccination provided | | 240 | 69.8 | 138 | 73.4 |
| Pneumococcal vaccine provided | | 19 | 5.6 | 23 | 12.2 |
| Pulmonary rehabilitation provided | | 17 | 4.9 | 42 | 22.3 |
| Inhaler technique assessed | | 56 | 16.3 | 116 | 61.7 |
| Inhalers prescribed | Salbutamol | 128 | 37.2 | 152 | 80.9 |
| | Ipratropium | 5 | 1.5 | 10 | 5.3 |
| | Salmeterol | 3 | 0.9 | 10 | 5.3 |
| | Fluticasone | 3 | 0.9 | 1 | 0.5 |
| | Budesonide | 0 | 0 | 0 | 0 |
| | Beclometasone | 20 | 5.8 | 13 | 6.9 |
| | Fluticasone/salmeterol | 33 | 9.7 | 70 | 37.2 |
| | Budesonide/formoterol | 12 | 3.5 | 23 | 12.2 |
| | Any of the above inhalers | 134 | 39 | 163 | 86.7 |
| Antibiotic rescue pack | | 10 | 2.9 | 43 | 22.9 |
| Prednisolone | | 51 | 14.8 | 96 | 51.1 |
| Clinical care score | <5 | 225 | 65.4 | 17 | 9 |
| | 5–9 | 83 | 24.1 | 73 | 38.8 |
| | ≥10 | 33 | 9.6 | 98 | 52.1 |
| | Median (IQR) | 3 | (2–5) | 10 | (7–12) |

*Number of participants who received the clinical assessment or intervention.
BMI, body mass index; CAT, COPD assessment test; COPD, chronic obstructive pulmonary disease; MRC, Medical Research Council.

## Clinical care of newly diagnosed COPD

Among patients with EHR data, clinical assessments that were commonly performed within the 2-year follow-up period included measurement of BMI (77.4%) and documentation of the MRC dyspnoea score (50.6%; table 4). Documentation of CAT score, disease severity, oxygen saturation, chest X-ray and depression screening, was infrequent. All aspects of clinical assessment were more commonly performed for patients diagnosed through usual care than for those by case finding.

Therapies that were commonly delivered included influenza vaccination (69.8% vs 73.4% for those diagnosed through case finding and usual care, respectively), smoking cessation counselling for current smokers

**Table 5** Clinical care of participants with follow-up questionnaire data who were newly diagnosed with COPD by case finding

| | | Listed on COPD register | | | |
| --- | --- | --- | --- | --- | --- |
| | | Yes (n=78) | | No (n=297) | |
| | | N* | (%) | N* | (%) |
| Informed about COPD diagnosis | | 69 | 88.5 | 52 | 17.5 |
| Annual review undertaken | | 65 | 83.3 | 103 | 34.7 |
| Spirometry undertaken | | 68 | 87.2 | 111 | 37.4 |
| Inhaler technique assessed | | 37 | 47.4 | 72 | 24.2 |
| Antibiotics prescribed | | 14 | 17.9 | 18 | 6.1 |
| Steroids prescribed | | 13 | 16.7 | 14 | 4.7 |
| Influenza vaccine offered | | 77 | 98.7 | 242 | 81.5 |
| Influenza vaccine received | | 67 | 85.9 | 207 | 69.7 |
| Pneumococcal vaccine offered | | 44 | 56.4 | 128 | 43.1 |
| Pneumococcal vaccine received | | 43 | 55.1 | 121 | 40.7 |
| Pulmonary rehabilitation offered | | 4 | 5.1 | 5 | 1.7 |
| Attended pulmonary rehabilitation | | 3 | 3.8 | 6 | 2 |
| Smoking cessation advice given | | 44 | 56.4 | 103 | 34.7 |
| Smoking cessation support offered† | | 34 | 43.6 | 60 | 20.2 |
| Inhalers prescribed | SABA | 45 | 57.7 | 87 | 29.3 |
| | SAMA | 21 | 26.9 | 18 | 6.1 |
| | ICS | 5 | 6.4 | 23 | 7.7 |
| | LABA | 0 | 0 | 5 | 1.7 |
| | LAMA | 7 | 9 | 1 | 0.3 |
| | ICS/LABA | 17 | 21.8 | 40 | 13.5 |
| | LABA/LAMA | 2 | 2.6 | 2 | 0.7 |
| | Any of the above | 58 | 74.4 | 104 | 35 |
| Care plan provided | | 62 | 79.5 | 39 | 13.1 |
| Clinical care score | <5 | 9 | 11.5 | 214 | 72.1 |
| | 5–9 | 56 | 71.8 | 76 | 25.6 |
| | ≥10 | 13 | 16.7 | 7 | 2.4 |
| | Median (IQR) | 8 | (6–9) | 3 | (2–5) |

*Number of participants self-reporting having received the clinical intervention.
†19/25 (76%) smokers listed on the COPD Quality and Outcomes Framework (QOF) register received smoking cessation support and 34/54 (63%) smokers not listed on the COPD QOF register had received this.
COPD, chronic obstructive pulmonary disease; ICS, inhaled corticosteroid; IQR, Interquartile range; LABA, long acting beta 2 agonist; LAMA, long acting muscarinic antagonist; SABA, short acting beta 2 antagonist; SAMA, short acting muscarinic antagonist.

(45.6% vs 73.9%), and prescription of inhalers (39.0% vs 86.7%). Prescription of NRT, pulmonary rehabilitation and pneumococcal vaccination were infrequent. The median clinical care score was significantly higher among participants who had been diagnosed by usual care than those by case finding (10 vs 3, respectively).

Among case-found patients with questionnaire data (n=375), those added to a COPD register (n=78 (20.8%)) were more likely to have been informed of their COPD diagnosis (88.5% vs 17.5%; table 5). They were also more likely to have received a number of clinical interventions, including a care plan (79.5% vs 13.1%), influenza and pneumococcal vaccination (85.9% vs 69.7%, and 55.1% vs 40.7%, respectively), and prescriptions of inhalers (74.4% vs 35.0%). Very few had been offered or referred to a pulmonary rehabilitation service irrespective of whether they had been added to a COPD register.

### Factors associated with higher levels of clinical care

Using EHR data, patients who had been added to a COPD register had a clinical care score 5 points higher than those who had not (adjusted mean difference 5.06, 95% CI 4.36 to 5.75; table 6). This was also found to a lesser extent for those with a higher number of comorbidities (adjusted mean difference 0.38 per additional comorbidity, 95% CI 0.11 to 0.65). These findings remained consistent in sensitivity analyses when data from the beginning of the trial were also included, although current smoking

**Table 6** Multilevel linear regression model assessing the association between the clinical care score* and participant characteristics, among those with electronic healthcare record data (n=467)

| Participant characteristics | aβ† | (95% CI) | P value |
|---|---|---|---|
| Age | 0.03 | (0.00 to 0.05) | 0.050 |
| Sex (male) | −0.24 | (−0.71 to 0.23) | 0.318 |
| Smoking status | | | |
| Ex-smoker | −0.13 | (−1.02 to 0.77) | 0.781 |
| Current smoker | 0.79 | (−0.13 to 1.71) | 0.094 |
| Comorbidities (n) | 0.38 | (0.11 to 0.65) | 0.007 |
| Case-found versus routinely diagnosed | −0.69 | (−1.44 to 0.07) | 0.076 |
| Listed on COPD register | 5.06 | (4.36 to 5.75) | <0.001 |

*Based on the clinical care of participants 2 years after the close of the TargetCOPD trial.
†Adjusted linear regression coefficient (this corresponds to the mean change in clinical care score for each unit rise in the independent variable), accounting for clustering by practice.
COPD, chronic obstructive pulmonary disease.

also became significantly associated with an increase in the clinical care score (1.09, 95% CI 0.21 to 1.97; online supplemental tables 5 and 6).

Similarly, among those with questionnaire data, patients who had been added to a COPD register had a higher clinical care score than those who had not (adjusted mean difference 3.48, 95% CI 2.81 to 4.15). This was also true to a lesser extent for those who had a higher CAT score (0.05 per unit rise in CAT score, 95% CI 0.01 to 0.08), and lower percentage of predicted $FEV_1$ (−0.02 per percentage rise, 95% CI −0.03 to −0.01; table 7).

**Table 7** Linear regression model assessing the association between the COPD clinical care score and participant characteristics, among those with case-found COPD and follow-up questionnaire data (n=293)

| Participant characteristics | aβ* | (95% CI) | P value |
|---|---|---|---|
| Age | 0.01 | (−0.03 to 0.04) | 0.719 |
| Sex (male) | −0.49 | (−1.06 to 0.09) | 0.096 |
| Smoking status | | | |
| Ex-smoker | 0.38 | (−0.49 to 1.24) | 0.393 |
| Current smoker | 0.74 | (−0.27 to 1.76) | 0.149 |
| FEV1 % predicted | −0.03 | (−0.04 to −0.01) | <0.001 |
| No. of comorbidities | 0.38 | (0.10 to 0.65) | 0.007 |
| CAT score | 0.05 | (0.01 to 0.08) | 0.005 |
| Listed on COPD register | 3.48 | (2.81 to 4.15) | <0.001 |

*Adjusted linear regression coefficient (this corresponds to the mean change in clinical care score for each unit rise in the independent variable).
CAT, COPD assessment test; COPD, chronic obstructive pulmonary disease; $FEV_1$, forced expiratory volume in 1 s.

## DISCUSSION
### Main findings
Despite being symptomatic and eligible for clinical care, only one in five patients newly diagnosed with COPD through targeted case finding in primary care had been added to a COPD register by the close of the TargetCOPD trial, compared with more than 90% of those diagnosed by usual care. Addition to the register was more likely among current and former smokers, and those with poorer lung function. Practice characteristics did not correlate with the percentage of case-found patients added to COPD registers.

The clinical care of COPD was significantly more comprehensive for patients who had been diagnosed by usual care than those by case finding, receiving on average seven more clinical assessments or interventions. This was mainly because patients diagnosed by usual care were significantly more likely to be added to a COPD register. In addition, patients with a higher number of comorbidities, worse CAT scores, poorer lung function and current smokers, were also more likely to receive a higher level of respiratory care.

Very few patients diagnosed by either approach had been offered or referred to pulmonary rehabilitation, irrespective of whether they had been added to a COPD register. Also, relatively few had been administered a pneumococcal vaccine or provided adequate smoking cessation support.

### Relationship to other studies
Only two published studies have evaluated the clinical care of patients newly diagnosed with COPD by case finding. Similar to our findings, these also showed that case-finding was not followed by adequate COPD management for most patients. One study based in the Netherlands examined community-dwelling frail patients aged 65 years and older with dyspnoea who had participated in a screening study for COPD and heart failure.[15] During 6 months of follow-up, only 13.7% (n=53) of the new cases of COPD had any changes made to respiratory drug prescriptions.

A large cluster RCT of COPD screening in the USA similarly found that respiratory-related clinical activity was limited following identification, with only 187 of 994 patients (19%) who screened positive for COPD receiving a respiratory intervention.[16] The study examined a limited number of clinical activities, which included referral for pulmonary function testing, referral to a respiratory specialist and new respiratory medication prescriptions. The likelihood of receiving this care was associated with prior visits for respiratory issues and previous prescriptions of respiratory medications.[17]

A qualitative study exploring the views of healthcare providers within the TargetCOPD trial on screening, suggested that poor knowledge, lower perceived priority and insufficient resources for COPD diagnosis and management were barriers to adequate COPD management in primary care.[18] A qualitative study with patients

also found that GPs often lack the time to engage in case finding, that accessing primary care appointments can be difficult, and that communication about a diagnosis can often be lacking.[19] In addition, patients may occasionally be in denial of their respiratory symptoms or may not prioritise this over other health issues. Case finding strategies will therefore need to address patient education on accessing health services more promptly for respiratory symptoms.

A recent literature review of COPD management in primary care found that there is significant variability in the provision of recommended treatments, with barriers including a lack of familiarity with clinical guidelines.[20] This is reflected in prescribing practices. An analysis of UK primary care prescribing data in 2014 among 24 957 patients found that COPD is often not managed according to the Global Initiative for Chronic Obstructive Lung Disease (GOLD) or NICE guidelines.[21] 18% of GOLD stage 2 patients had received no treatment despite having symptoms, and a significant proportion had received inhaled corticosteroids irrespective of their disease severity and exacerbation history. Similar findings of inconsistent prescribing were found in a study assessing the management of COPD in a primary care clinic in the USA.[22]

More recently, the national COPD primary care audit in Wales found significant shortcomings in the clinical management of patients with COPD.[23] Only 12.5% of smokers had received smoking cessation support, 34.0% had not received an influenza vaccination and half of patients with MRC dyspnoea grades 3–5 had not been referred for pulmonary rehabilitation. A study of COPD care in community pharmacies in Belgium similarly found low rates of influenza vaccination in patients younger than 65 years, poor medication adherence and poor inhaler technique, among a significant proportion of COPD patients.[24] Findings from the Continuing to Confront COPD Survey suggested that there are likely to be significant shortcomings in the provision of guideline-recommended treatments among both primary and secondary care clinicians internationally.[25]

The UK National Screening Committee recommend that clinical service provision and patient outcomes should be optimised in all healthcare providers prior to participation in a screening programme.[5] These widespread gaps in care provision will need to be addressed before recommendations for targeted case finding can be made.

### Limitations
Limitations include the unavailability of data on all trial participants. Patient questionnaires were only available for 375 case-found patients and EHR data on clinical care for a subset of participants (344 out of 857 (40.1%) diagnosed by case finding and 188 out of 764 (24.6%) by usual care). However, the demographic and clinical characteristics of patients with and without relevant data were broadly similar, suggesting that the findings are likely

to apply to the full study population (see online supplemental table 3). There was also limited data on disease severity, particularly in the usual care arm of the Target-COPD trial. However, our analyses adjusted for measures of disease severity such as percentage of predicted $FEV_1$, CAT score and number of comorbidities.

EHR data quality was highly dependent on clinical coding by participating practices. These were routinely collected health service data and were not specifically recorded for research purposes. Inadequate recording could lead to an underestimation of the level of clinical care provided. Furthermore, a significant amount of smoking cessation support in England is provided in community pharmacies, which would not necessarily be captured in GP records. However, self-reported questionnaire data on smoking cessation support were also collected from a subset of patients, and had broadly similar findings.

The clinical care score represents a relatively crude measure of overall respiratory care and individual components of the score were not weighted for their relative importance. Some of the components reflect the management of COPD exacerbations, which may have differed in their incidence between case-found and routinely diagnosed patients. However, there are currently no validated methods or scores for quantifying overall levels of care for COPD and we therefore chose what we considered a reasonable and pragmatic approach. Finally, the primary outcome of addition to a COPD register is specific to the UK-context. However, our findings do suggest that COPD registries play an important role in supporting COPD management and could be encouraged elsewhere.

### Implications for practice, policy and research
Our findings suggest that COPD case finding in primary care is unlikely to result in improvements to clinical care. In the context of UK-based primary care and similar health systems, it should not be implemented in the absence of care pathways[7] to ensure that case-found patients are promptly added to primary care COPD registers and receive appropriate management. Further trials investigating the effectiveness of COPD case finding are unlikely to be ethical in the absence of such pathways of care. Encouragingly, we identified one trial protocol that aims to evaluate COPD case finding in conjunction with an integrated care pathway in low-income and middle-income countries[26] but more such trials will be needed to make firm recommendations.

In England, including patients on COPD primary care registers is associated with financial reimbursement through the Quality and Outcomes Framework.[27] This requires a number of care quality indicators, such as performing diagnostic spirometry and providing annual influenza vaccination, to be documented in electronic healthcare records. This may at least partly explain why case-found patients who had been added to a disease register received significantly higher levels of COPD-related care than patients who had not been

added. New indicators have recently been added to this scheme, including referral to a pulmonary rehabilitation programme for patients with MRC dyspnoea ≥3. This could potentially improve the levels of care provided to these patients.

Even in the presence of robust care pathways, further research and modelling will be needed to assess whether there is sufficient health service capacity to meet the demands of the additional cases of COPD that would be detected through targeted case finding. Most importantly, research is needed to empirically evaluate whether targeted case finding improves both short and long-term clinical outcomes and healthcare costs compared with usual care.

## Conclusions

Only a minority of patients with case-found COPD in primary care are likely to receive adequate levels of clinical care. Case finding is only likely to improve clinical care if patients with newly identified disease are promptly added to a primary care COPD register. This appears to be more likely to occur for patients who are current or former smokers, and have more severely impaired lung function. Further research is needed to model the impact of case finding on health service capacity, and to evaluate its effectiveness on clinical outcomes and costs.

**Contributors** REJ, PA and DAF were principal investigators on the TargetCOPD trial and were responsible for the follow-up of the trial. SH developed the protocol with input from REJ, PA and APD. SH conducted the analysis with input from REJ, PA and AJS. APD and AE coordinated the collection of EHR follow-up data. KR developed the follow-up questionnaires with input from REJ, PA, AE and SH. SH wrote the manuscript with input from all coauthors. All authors have contributed to and approved the final version.

**Funding** This paper presents independent research funded by the National Institute for Health Research (NIHR) under its Programme Grants for Applied Research Programme (Grant Reference Number RP-PG-0109-10061) and by the NIHR Birmingham Biomedical Research Centre at the University Hospitals Birmingham NHS Foundation Trust and the University of Birmingham. The views expressed are those of the author(s) and not necessarily those of the NIHR, the NHS or the Department of Health and Social Care. TargetCOPD is part of The Birmingham Lung Improvement StudieS—BLISS.

**Competing interests** PA, REJ and DAF were principle investigators for the Birmingham Lung Improvement StudieS (BLISS) programme, funded by the National Institute for Health Research (NIHR) under its Programme Grants for Applied Research (grant reference number RP-PG-0109-10061). We report grants from various NIHR studies. PA is the Chair of the NIHR Public Health Research (PHR) Funding committee.

**Patient consent for publication** Not required.

**Ethics approval** The Solihull Research Ethics Committee provided ethical approval for the TargetCOPD trial (IRAS, reference 11/WM/0403).

**Provenance and peer review** Not commissioned; externally peer reviewed.

**Data availability statement** Data are available upon reasonable request. All data requests should be submitted to authors REJ and PA for consideration. Access to anonymised data may be granted following review.

**ORCID iDs**
Shamil Haroon http://orcid.org/0000-0002-0096-1413
Peymane Adab http://orcid.org/0000-0001-9087-3945
Andrew P Dickens http://orcid.org/0000-0002-7591-8129
Alice J Sitch http://orcid.org/0000-0001-7727-4497
Kiran Rai http://orcid.org/0000-0002-3250-0275

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
