## [Reviewer comments · BMJ Open]

ARTICLE DETAILS

TITLE (PROVISIONAL)	The impact of COPD case finding on clinical care: a prospective analysis of the TargetCOPD trial
AUTHORS	Haroon, Shamil; Adab, Peymane; Dickens, Andrew; Sitch, Alice; Rai, Kiran; Enocson, Alexandra; Fitzmaurice, David; Jordan, Rachel

VERSION 1 – REVIEW

REVIEWER	David M MANNINO University of Kentucky Glaxo Smith Kline USA
REVIEW RETURNED	19-Apr-2020

GENERAL COMMENTS	Well written paper describing the clinical outcomes of the TargetCOPD casefinding trial. It was a bit disappointing in that the cases detected via case finding were much less likely to be added to the COPD registry than those detected via "usual care" (but not terribly surprising) Major comments - None Other comments - This paper is fine as is. It seems a follow-up analysis (for another paper) would be to look at other key outcomes in the case found group relative to the usual care group (i.e.- looking at hospitalizations, mortality, etc) to see if there is any difference between the "usual care" , the case found added (who are presumably ealier in the disease process), and the case found not added.
---

REVIEWER	Barbara Yawn, MD MSc University of Minnesota, USA
	Am investigator in ongoing study of COPD screening/case finding in primary care practices in the USA and a COPD registry study in primary care practices in the USA and the Co-PI of teh COPD Foundation's Patient Powered Research Network (patient registry of individuals with self-reported COPD).
REVIEW RETURNED	27-Apr-2020

GENERAL COMMENTS	An important study of outcomes from case finding for COPD. The overall conclusions do not appear to consider whether physicians actually received and reviewed the information from the case finding. Has this been studied in one of the mentioned qualitative analyses? Physicians receive many notices for many things, what did you do to make the results from this case finding noticeable or to "stand out"? Did you also report results to the patients?
---

	Specific issues--what did you only send questionnaires to the case found patients? The primary outcome of addition to a registry is important for UK but not translatable to other countries. The secondary outcomes seemed to be assessed only from end of trial to 2 years later with comment that sensitivity analyses were done from 3 years but did not see those results. Weren't many of the things done immediately? Agree that the clinical care is a composite that does not meet clinical importance metrics. Page 9--how did you do a model with survey results if usual care patients were not surveyed? Page 10--response rate was 44% and should be stated as such. The abstract comments on lower FEV1 results or impact on outcomes but I could not find the FEV1 data reported in the manuscript or the tables. In summary--important and recommendations to include patients in care team and educational programs is very important but what can you do to get the attention of the physicians and how many of the case found patients were candidates for immediate COPD clinical interventions?
--	--

VERSION 1 – AUTHOR RESPONSE

Reviewer: 1

Reviewer Name: David M MANNINO

Institution and Country: University of Kentucky Glaxo Smith Kline USA Please state any competing interests or state 'None declared': None Declared

Please leave your comments for the authors below Well written paper describing the clinical outcomes of the TargetCOPD casefinding trial. It was a bit disappointing in that the cases detected via case finding were much less likely to be added to the COPD registry than those detected via "usual care" (but not terribly surprising)

Major comments - None

Other comments - This paper is fine as is. It seems a follow-up analysis (for another paper) would be to look at other key outcomes in the case found group relative to the usual care group (i.e.- looking at hospitalizations, mortality, etc) to see if there is any difference between the "usual care" , the case found added (who are presumably ealier in the disease process), and the case found not added.

Our response: Many thanks for reviewing our paper; this is indeed the subject of another paper under development.

Reviewer: 2

Reviewer Name: Barbara Yawn, MD MSc

Institution and Country: University of Minnesota, USA Please state any competing interests or state 'None declared': Am investigator in ongoing study of COPD screening/case finding in primary care practices in the USA and a COPD registry study in primary care practices in the USA and the Co-PI of teh COPD Foundation's Patient Powered Research Network (patient registry of individuals with self-reported COPD).

Please leave your comments for the authors below An important study of outcomes from case finding for COPD. The overall conclusions do not appear to consider whether physicians actually received and reviewed the information from the case finding. Has this been studied in one of the mentioned qualitative analyses? Physicians receive many notices for many things, what did you do to make the results from this case finding noticeable or to "stand out"? Did you also report results to the patients?

Our response: Many thanks for reviewing our manuscript. The study team sent GPs the spirometry results for all participants found to have spirometry-confirmed airflow obstruction during the trial. A linked qualitative study of GPs involved in the trial (reference 19 in the paper) revealed one of the key themes to emerge was that primary care services were already at capacity managing existing COPD patients and a lack of resources was a barrier to implementing case finding, which may explain this lack of action. A second linked qualitative study (reference 20 in paper) with patients found as one of its themes that GPs often lack the time to engage in case finding, which we report on page 15 of the manuscript. We did not report the spirometry results to patients directly as our study protocol, in accordance with the ethical approvals received, required the information to be passed to GPs who were then responsible for conveying that information to patients according to clinical need.

Specific issues--what did you only send questionnaires to the case found patients?

Our response: Due to the study design, patients in the routine arm of the trial did not need to individually consent to participate, as the yield was obtained from aggregate-level COPD registers at the end of the study period. As we did not seek consent of routine arm patients, we were unable to disseminate questionnaires to this group.

The primary outcome of addition to a registry is important for UK but not translatable to other countries.

Our response: We agree that the primary outcome of addition to a registry is specific to the UK-context. However, our findings suggest that this approach is likely to be beneficial for supporting COPD management in other countries with similar health systems. The register is a proxy for noting the diagnosis in GP records, prompting appropriate management. We have added the following statement to the section on study limitations to qualify this: "The primary outcome of addition to a COPD register is specific to the UK-context. However, our findings do suggest that COPD registries may play an important role in supporting COPD management."

The secondary outcomes seemed to be assessed only from end of trial to 2 years later with comment that sensitivity analyses were done from 3 years but did not see those results. Weren't many of the things done immediately?

Our response: The secondary outcomes were assessed primarily from the end of the trial up to 2 years of follow-up. We recognise that some aspects of clinical care may have taken place more immediately, but two years allowed time for all administrative processes to be completed, time for the patient to have been reviewed and for any further assessments and treatments to have initiated. We did also perform sensitivity analyses including data from the beginning of the trial up to three years of follow-up (page 9 of the manuscript), which included interventions that could have been delivered immediately after the diagnosis. The findings were comparable between both analyses and we have provided the results of the sensitivity analyses in supplementary tables 5 and 6 (which we have indicated in the manuscript on page 13).

Agree that the clinical care is a composite that does not meet clinical importance metrics.

Page 9--how did you do a model with survey results if usual care patients were not surveyed?

Our response: The study included several different statistical models, some using EHR data and some using data from questionnaires about clinical management. The latter were only sent to case-found patients as discussed above. We wished to investigate what factors were associated with the degree of clinical care received by participants with case found disease. This would not have applied to patients in the usual care arm who would not have undergone case finding. We have clarified this in the statistical methods and the caption for table 7.

Page 10--response rate was 44% and should be stated as such.

Our response: We have now added the response rate to the results.

The abstract comments on lower FEV1 results or impact on outcomes but I could not find the FEV1 data reported in the manuscript or the tables.

Our response: FEV1 data is provided in supplementary tables 3 and 4 and the results of the regression models with FEV1 as a covariate are provided in tables 2 and 7.

In summary--important and recommendations to include patients in care team and educational programs is very important but what can you do to get the attention of the physicians and how many of the case found patients were candidates for immediate COPD clinical interventions?

Our response: While the majority of participants with case-found COPD had mild disease, all were symptomatic (as described on page 8 of the manuscript) and therefore were eligible for at least some aspects of COPD care according to NICE guidelines. For example, 30% of case-found patients were current smokers and would benefit from smoking cessation therapy, and all were symptomatic and would therefore be eligible for inhaled therapies. All would also benefit from self-management support. We have now emphasised this in the discussion under the main findings. We have also emphasised the importance of care pathways for case-found patients under "Implications for practice, policy and research".

FORMATTING AMENDMENTS (if any)

Required amendments will be listed here; please include these changes in your revised version:

- The author " Andrew P. Dickens" in your main document is registered as " Dickens, Andy" in ScholarOne. Please ensure that the author has same registered name.

Our response: We believe this discrepancy has now been resolved, as it was identified in relation to another recent manuscript submission, and Andrew Dickens subsequently updated his ScholarOne account. If this is still an issue, please advise us how to edit the ScholarOne account accordingly.

- Please re-upload your supplementary files in PDF format.

Our response: The supplementary file has now been re-uploaded in PDF format.

- Figure/s should not be embedded

Please remove all your figures in your main document and upload each of them separately under file designation 'Image' (except tables and please ensure that figures are in better quality or not pixelated when zoomed in).

They can be in TIFF, JPG or PDF format. Make sure that they have a resolution of at least 300 dpi and at least 90mm x 90mm of width. Figures in document, excel and powerpoint format are not

acceptable.

Our response: The figure has been removed from the main document and uploaded as a JPG file.

- Table citation missing

The in-text citation for 'table 7' is missing. Please provide the missing citation and ensure that all citations of tables are in ascending order.

Our response: The in-text citation for table 7 should now be visible.

VERSION 2 – REVIEW

REVIEWER	Barbara P Yawn, MD MSc University of Minnesota Currently working on COPD screening study in the US.
REVIEW RETURNED	08-Jul-2020

GENERAL COMMENTS	The authors have addressed my concerns.
---